# AMD-FV: Adaptive margin loss and dual path network+ for deep face verification

**Zeeshan Ahmed Khan**[1], **Waqar Ahmed**[2], **Panos Liatsis**[2]*

1 Department of Computer Science, School of Informatics, Xiamen University, Fujian, China,
2 Department of Computer Science, Khalifa University, Abu Dhabi, United Arab Emirates

* panos.liatsis@ku.ac.ae

**Data availability statement:** All Labeled Faces in the Wild (LFW) files are available from the UMass database (https://vis-www.cs.umass.edu/lfw/). All Megaface files are available from the University

## Abstract

Face verification is important in a variety of applications, for instance, access control, surveillance, and identification. Existing methods often struggle with the challenges of dataset imbalance and manual hyperparameter tuning. To address this, we propose the Adaptive Margin Loss and Dual Path Network+ (AMD-FV) for deep face verification. Two innovations are introduced, namely, Adaptive Margin Loss (AML) and Dual Path Network+ (DPN+). AML aims at automating the selection of margin and scale hyperparameters in large margin loss functions, thus, eliminating the need for manual tuning. Input dissimilarity information is used to estimate the margin, while the scale parameter is computed using the number of classes and AML's range. Next, DPN+ enhances the original Dual Path Network by redesigning the first block with a series of 3x3 convolutions, batch normalization, and ReLU activations, leveraging shared connections across layers, leading to increases in spatial resolution and computational cost efficiency, while maximizing the use of discriminative features. We present comprehensive experiments on five diverse face verification datasets (LFW, Megaface, IJB-B, CALFW, and CPLFW) to demonstrate the effectiveness of the proposed approach. The results show that AMD-FV outperforms state-of-the-art methods, achieving a verification accuracy of 99.75% on LFW, improving the True Acceptance Rate by 6% on IJB-B at a False Acceptance Rate of 0.001, compared to VGGFace2, and attaining a Rank-1 identification score of 92.16% on Megaface, surpassing the CosFace model by 9.44%.

## Introduction

Traditional methods of human identification relying on credentials (e.g., PINs, identification documents, etc) are unable to meet the growing demands for stringent security in applications such as access control, surveillance, and identification. Alternatively, biometrics (e.g., Fingerprint, Iris, Vein, Voice, Face, etc) gained significant attention over the past two decades as an alternative solution. Face verification (FV) as a biometric authentication method is particularly significant. It offers several advantages over other biometric traits. For instance, it is non-intrusive and easy to use due to the widespread availability of cameras in a variety of devices and environments. It is widely used in several applications, including security systems and mobile devices, to provide secure access to personal devices and sensitive information.

of Washington database (https://megaface.
cs.washington.edu/dataset/download.html). All
IARPA Janus 215 Benchmark B (IJB-B) files are
available from the NIST database (https://www.
nist.gov/programs-projects/face-challenges). All
Cross-Age LFW (CALFW) files are available from
the whdeng database (http://www.whdeng.cn/
CALFW/index.html?reload=true). All Cross-Pose
LFW files are available from the whdeng
database (http://www.whdeng.cn/CPLFW/
index.html?reload=true).

**Funding:** The author(s) received no specific
funding for this work.

**Competing interests:** The authors have
declared that no competing interests exist.

Discriminative feature extraction from image/video data is of paramount importance in face verification. Early studies [1–4] focused on utilizing hand-crafted feature extraction methods. Later, deep learning based methods revolutionized feature extraction by autonomously learning intricate patterns and representations from data automatically [5–7]. In particular, network architecture and loss function play crucial and complementary roles in feature extraction. The architecture defines the network's structure and capacity to learn representations, while the loss function guides the learning process towards extracting meaningful and discriminative features, relevant to the task at hand. The synergy between these components is essential for achieving optimal performance.

From a loss function design perspective, SphereFace [8] introduces a multiplicative angular margin loss function. CosFace [9] demonstrates the effectiveness of subtracting a margin rather than multiplying it. ArcFace [10] proposes an additive margin approach, which facilitates the learning of discriminative features. However, the limitation of SphereFace [8] is that its performance is sensitive to hyperparameters, particularly the choice of the angular margin parameter. Indeed, improper tuning of this parameter may lead to suboptimal feature discrimination. CosFace [9] exhibits computational complexity, especially when dealing with large-scale datasets. Specifically, the computation of the cosine similarity with respect to each class centroid during training can become computationally expensive, potentially affecting training times and resource efficiency. ArcFace [10] relies on hyperparameters that necessitate manual selection, involving repeated trial and error, which is time-consuming. Thus, on one hand, the hyperparameters of angular margin loss functions offer leverage for tuning deep learning architectures to achieve improved results. On the other hand, improper selection of these hyperparameters may substantially impact accuracy.

To the best of our knowledge, to date, AdaptiveFace [11] and AdaCos [12] are the primary methods offering automated hyperparameter selection for large margin loss frameworks. AdaptiveFace [11] estimates the margin hyperparameter, while AdaCos [12] focuses on the scale hyperparameter. Several related approaches address various aspects of hyperparameter optimization. For example, HAMFace [13] introduces a Hardness Adaptive Margin Loss, applying larger margins to challenging positive samples and using a control hyperparameter to jointly manage the margin and scale hyperparameters. The study by Gim et al., [14] incorporates Noise Direction Regularization (NDR), leveraging the $L_2$ norm of the features as a regularizer and introducing a hyperparameter to improve performance. CoReFace [15] further advances the field by employing an adaptive margin-based contrastive loss, which utilizes an exponential moving average for the margin, while relying on manual scale tuning. To overcome the limitations of existing methods, this study proposes AML, a novel framework that automates the selection of both margin and scale hyperparameters. This estimates the margin using input dissimilarity information derived from the cross-product of the feature and weight vectors, while the scale hyperparameter is determined based on the number of classes and the effective range of AML (more details in Adaptive margin loss).

From a network architecture design perspective, the Deep Residual Network (ResNet) [16] introduced skip connections, where each micro-block (residual function) includes a residual path. This path element-wise adds input features to the output of the same micro-block, defining it as a residual unit. ResNet evolved into various architectures such as Inception-Resnet [17] and ResNeXt [18], each characterized by changes in their structural designs. Later, [19] proposed the Dense Convolutional Network (DenseNet), which uses dense connections, with each layer receiving inputs from all preceding layers and passing its own feature maps to all subsequent layers, thus enhancing information flow and feature reuse. Chen et al., [20] explore the relationship between residual networks and densely connected networks, demonstrating that residual networks exhibit characteristics of densely connected networks

when connections are shared across layers. They proposed the Dual Path Network (DPN), which leverages the residual path for implicit feature reuse and continuous exploration of new features through the densely connected path.

Several studies [16–20] rely on downsampling in the network architecture, however, this often leads to significant information loss. García et al., [21] highlighted the challenges of preserving diagnostic features in medical images, proposing feature-preserving techniques, which, despite improvements, were unable to completely prevent degradation. Similarly, Sedlar et al., [22] noted that inappropriate downsampling in genomic signal processing obscured critical patterns, advocating for tailored rules to optimize the balance between data reduction and feature retention. Zhao et al., [23] introduced random shifting to minimize systematic information loss in CNN downsampling layers, enhancing feature detail preservation. Kuen et al., [24] proposed stochastic downsampling to improve network regularization and enable cost-adjustable inference, however, this introduces variability, which could affect feature consistency. To address this limitation, we introduce the Dual Path Network+ with a carefully redesigned Block-1, which includes a series of 3x3 convolutions, batch normalization, and ReLU activation functions to enhance spatial resolution and reduce computational costs. Moreover, DPN+ maintains shared connections across layers to maximize the utilization of discriminative features, thereby enhancing both performance and efficacy. More details can be found in Dual path network+.

In summary, existing approaches suffer due to the requirement for manual hyperparameter selection, which significantly impacts on their performance, or alternatively, facing computational challenges, when it comes to dealing with large-scale, real-world datasets. Moreover, due to the use of downsampling in the CNN architectures, there is significant information loss, reducing the discriminating ability of the underlying features. Fig 1 provides an overview of the proposed AMD-FV framework, incorporating Dual Path Network+ and Adaptive Margin Loss. This achieves state-of-the-art performance in face verification tasks across five diverse datasets, varying in volume and complexity.

The contributions of this work are as follows:

- Adaptive estimation of the margin hyperparameter in large margin loss is achieved. The proposed method utilizes the dissimilarity of input data, estimated via the cross product of the feature and weight vectors, to adaptively adjust the marginalization of features.
- The scale hyperparameter of the large margin loss is automatically selected by leveraging the number of the classes and AML range. This adaptive approach scales the marginalized features for improved classification, enhancing the effectiveness of the model.
- The Dual Path Network+ is introduced with a redesigned Block-1, which enhances spatial resolution and reduces computational costs. This block incorporates a series of 3x3 convolutions, batch normalization, and ReLU activation functions. Moreover, the architecture leverages shared connections across layers to maximize the utilization of discriminative features, ultimately improving both performance and efficacy.

The remainder of the paper is organized as follows. In Section Related works, related works in the literature are discussed and analyzed. Section Materials and methods describes the details of the proposed method. Section Experiments illustrates the experimental setup, presents the details of the utilized datasets and reports on the implementation details. Section Results presents a thorough ablation study, followed by a performance comparison with related works, and a discussion of the key findings. Finally, conclusions and future work are drawn in Section Conclusions.

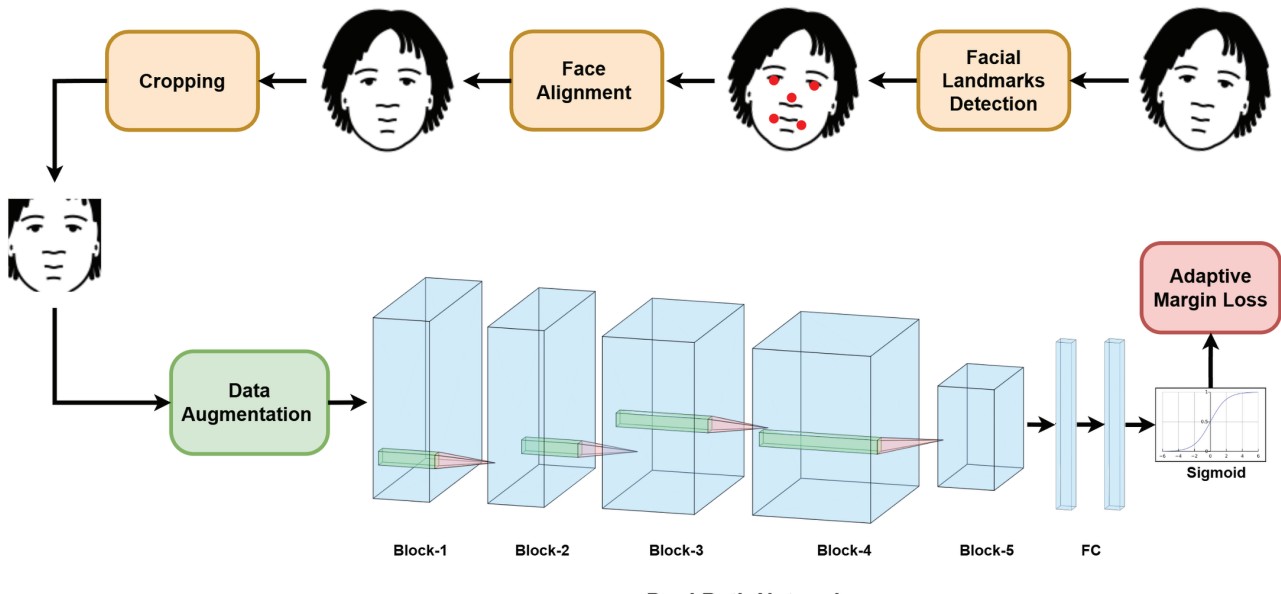

**Fig 1. Overview of the AMD-FV method.** Adaptive Margin Loss is used in adaptive estimation of the margin and scale hyperparameters. The Dual Path Network+ offers increased spatial resolution at reduced computational costs, via shared connections across layers to maximize the utilization of discriminative features. Data preprocessing includes facial landmarks detection, face alignment using five key facial points (i.e., nose, eyes, and corners of the lips), and cropping. Next, the cropped image undergoes data augmentation to provide the network with various versions of the same input, facilitating the learning of discriminative features.

## Related works

Margin-based loss functions, such as SphereFace [8], CosFace [9], and ArcFace [10], demonstrated significant success in face verification tasks. The hyperparameters associated with these loss functions, often manually chosen, play a crucial role on the performance of the deep learning model. Indeed, suboptimal or incorrect selection of these hyperparameters can significantly affect their properties and, subsequently, model performance. Typically, determining these parameters involves a time-consuming and laborious trial-and-error process, the success of which is limited by the choice of initial parameters. To address this issue, researchers have proposed data-driven methods to estimate these hyperparameters. For instance, Liu et al., [11] argue that face data typically consists of large-scale classes and samples, emphasizing that selecting appropriate classes and sample numbers for training helps the model achieve better generalization. They proposed the use of hard prototype mining and introduced the AdaptiveFace model to estimate the associated hyperparameters. However, the computational overhead of learning individual margins for each class thoroughly considered. In the case of large-scale datasets, this could potentially lead to significant increases in training time and memory requirements. Moreover, as demonstrated in the experiments, the performance of the method is sensitive to the choice of the hyperparameter controlling the strength of the margin constraint. Zhang et al., [12] propose the Adacos model to tackle the challenge of hyperparameter selection in face verification tasks. However, their approach presents a number of challenges. First, it assumed that during training, the sum of logits for non-corresponding classes remains almost constant. Beyond empirical evidence, the theoretical justification for this assumption, crucial in deriving the adaptive scale parameter, is not proven. Moreover, the analysis of probability curves is based on simplified models, which may

not fully capture the complexity of real-world face recognition scenarios, e.g., class imbalance and samples of varying difficulty.

Zhang et al., [25], propose diversity regularization by training the margin hyperparameter and constraining it within the range of 0.5 and $\pi$. While the authors automatically train the margin hyperparameter, they introduce two additional manual parameters in terms of the range. Moreover, there is insufficient analysis of the impact of the balance hyperparameter across different datasets and network architectures, and the computational overhead associated with the calculation of adaptive margins and diversity regularization. RamFace [26] addresses racial bias by proposing an adaptive margin loss based on intra-identity compactness using multitask learning. Two loss functions, i.e., race adaptive margin-based loss and race classification loss, are introduced, however, the additional hyperparameter $\lambda$ is used to balance them. There is insufficient consideration on the appropriate choice of this hyperparameter, and its impact on model performance. In addition, the manuscript neither provides formal guarantees on the fairness properties of the proposed method, nor does it discuss in sufficient detail the potential trade-offs between overall accuracy and fairness across different racial groups. EnhanceFace [27], proposes adaptive weighted softmax loss by segregating hard samples into semi-hard and harder samples. The loss function adjusts the corresponding weights during training. The authors introduce the enhance rate $\omega$, used to adjust the effects of semi-hard samples. However, the selection of this parameter appears arbitrary, as there is lack of justification for the chosen range and the step size. JAMsFace [28] proposes a joint adaptive margin loss function, which learns class-related margins by incorporating margin penalties within the cosine angle of a loss function. These models mitigate the problem to some extent. However, they only provide a means of estimating a single hyperparameter, i.e., margin, and moreover, there is a lack of a clear formulation on how this is calculated based on class distribution. In contrast, the proposed AML approach estimates both the scale and margin hyperparameters.

In the context of CNN architectures, García et al., [21] discussed the difficulty of preserving diagnostic features in medical images during downsampling. He et al., [29] introduced the Residual Network (ResNet), which utilizes skip connections to enable very deep networks, addressing optimization challenges. However, the use of downsampling in ResNet leads to significant information loss. Sedlar et al., [22] identified similar challenges in genomic signal processing, calling for strategies that balance data reduction with feature retention. Zhao et al., [23] proposed random shifting to minimize systematic information loss during downsampling. In contrast, Huang et al., [30] introduced the Dense Convolutional Network (DenseNet), which uses skip connections to concatenate features from earlier layers, but at the expense of high computational complexity. Chen et al., [20] demonstrated that ResNet reuses features via residual paths, while DenseNet explores new features through densely connected paths. Building on this, Chen et al., [20] proposed the Dual Path Network, which combines the strengths of both ResNet and DenseNet, although it still suffers from information loss due to its downsampling approach. The proposed Dual Path Network+ is an enhancement to the original DPN, improving spatial resolution and computational efficiency, while leveraging shared connections across layers to maximize feature utilization.

## Materials and methods

### Background

Angular margin loss function extends Cross-Entropy Softmax Loss (CESL) through the introduction of hyperparameters, which is given as follows:

$$L_{(Softmax)} = -\frac{1}{N} \sum_{i=1}^{N} log \frac{e^{W_{y_i}^{(T)} x_i + b_{y_i}}}{\sum_{j=1}^{n} e^{W_j^{(T)} x_i + b_j}} \tag{1}$$

where $x_i$ denotes the embedded feature vector of the $i^{th}$ sample of the $y_i$ class, $W_{y_i}, W_j$ are the $y_i^{th}$ and $j^{th}$ columns of the weights, respectively, whereas $b_{y_i}, b_j$ are the respective biases. $N$ and $n$ denote the batch size and the number of classes, respectively.

The CosFace model [9] proposes a Large Margin Cosine Loss (LMCS) and removes the radial variation of the features and weights [9]. The last fully connected (FC) layer operates on the normalized features $x_i$ and the normalized weights $W_j$. The norm of $\|W_j\|$ is set to 1 and the norm of $\|x_i\|$ is replaced by the hyperparameter $s$. Hence the activation of the last FC layer is given by $W_j^{(T)} x_i + b = s \cos \theta_j$. LMCL is given by:

$$L_c = -\frac{1}{N} \sum_{i=1}^{N} log \frac{e^{s*(\cos \theta_{y_i} - \beta)}}{e^{s*(\cos \theta_{y_i} - \beta)} + \sum_{j=1,j\neq y_i}^{n} e^{s*(\cos \theta_i)}} \tag{2}$$

where $s$ and $\beta$ denote the scale and margin hyperparameters respectively.

## Adaptive margin loss

In this section, we introduce the proposed angular margin-based loss function, AML, which streamlines the selection process for the scale hyperparameter $s$ and margin hyperparameter $\beta$ in the CosFace loss function. The inspiration for this approach stems from the observation that humans naturally take into account dissimilarities when comparing images. For instance, when presented with two images, e.g., one of a male and the other of a female, individuals instinctively focus on their differences rather than similarities to distinguish between them. This observation suggests that the concept of dissimilarity could serve as a basis for matching. In contrast, traditional neural networks typically measure the degree of similarity between the presented input patterns and the network weights.

In this research, we leverage the concept of dissimilarity akin to human visual perception to introduce a novel methodology for the automated estimation of the margin hyperparameter. The dissimilarity of a batch of images is assessed based on the magnitude of the cross product, subsequently subtracting it from the feature map. While the proposed approach adopts the architecture of CosFace, it differs significantly, as CosFace relies on a manually selected margin value. For the automated estimation of the scale hyperparameter, we consider the product of the number of classes within the training dataset and the range of the loss function. The proposed AML loss function operates on the normalized features $x_i$ and the normalized weights $W_j$ of the last FC layer. It automatically estimates both the margin and the scale hyperparameters.

For the adaptive estimation of margin hyperparameter $m$, we estimate the mean dissimilarity of the feature vectors $x_i$ and weight vectors $W_j^{(T)}$ in the given batch of images by using the magnitude of the cross product (see A. Magnitude of Cross Product for more details). We consider the mean distance as the probable average dissimilarity of the batch $n$ and estimate the margin hyperparameter $m$ as follows:

$$m = \frac{1}{n} \sum_{i=1}^{n} |W_j^{(T)} \times x_i| \tag{3}$$

For the adaptive estimation of the scale hyperparameter $s$, the logarithm of the number of classes and the maximum range is used as follows:

$$s = log(N)\, e^R \tag{4}$$

where $N$ is the number of classes, and $R$ is the maximum range of AML (see B. Range of AML for more details). The above choice is motivated by [31], where it is argued that the logarithmic function is effective in scaling down large values, thus facilitating easier mathematical operations. Conversely, the exponential function is utilized to provide leverage, enabling the estimation of the rate of change with precision within small increments. In our scenario, we deal with a large dataset (MS1Mv2) comprising 85,000 classes, with the activation of the last fully connected layer being in the range of 0 to 1. Classifying 85,000 classes within the spectrum of [0,1] presents a considerable challenge. Therefore, the proposed method works by appropriately scaling the marginalized features using the logarithmic function to mitigate the impact of the large number of classes. Subsequently, we utilize the exponential function to emphasize the small range (i.e., 0 to 1) associated with the activation of the last FC layer. The steps of the proposed AML are outlined in Algorithm 1. The Adaptive Margin Loss function $L_{AML}$ with the automatically selected margin hyperparameter $m$ and scale hyperparameter $s$ is as follows:

$$L_{AML} = -\frac{1}{N}\sum_{i=1}^{N} log \frac{e^{s*(cos\,\theta_{y_i}-m)}}{\sum_{j=1}^{n} e^{s*(cos\,\theta_j-m)}} \tag{5}$$

**Algorithm 1. Adaptive Margin Loss Algorithm.**

**Require:** Feature map $x_i$ and weights $W_j$ of the last fully connected layer.

**Ensure:** The updated loss via the estimated scaled and marginalized parameters.

1: Normalize the feature map $x_i$ in the batch.
2: Normalize the weights $W_j$ in the batch.
3: Update the last FC layer through the normalized feature map and the normalized weights.
4: Estimate the margin hyperparameter value via Eq (3).
5: Estimate the scale hyperparameter value via Eq (4), where N is the number of classes, and R is the maximum range of AML (see B. Range of AML for details).
6: Update the output of the FC layer by multiplying the scale hyperparameter and subtracting the margin hyperparameter.
7: Update the loss using Eq (5).

## Dual path network+

As discussed in [24], [22], [21], and [32], downsampling in CNN represents a trade-off between low-level spatial information and richer high-level semantic information. Therefore, in the proposed method, to prevent information loss, we refrain from downsampling and instead, leverage the low-level features in the first convolutional stage of DPN. This approach leads to the development of the proposed Dual Path Network+, which is a 68-layer

convolutional neural network inspired by [20]. The overall neural network structure of DPN+ is presented in Table 1.

DPN+ is constructed by stacking five stages of convolution, as outlined in Table 1. The structure of the first block comprises a series of three convolutional layers, where each layer consists of a $3 \times 3$ convolution with a stride of 1, followed by Batch Normalization [33] and ReLU activation [34]. This initial stage leverages low-level feature information as the foundation for generating highly discriminative features. In the first block of DPN+, information loss due to downsampling is mitigated, and low-level feature information is enhanced through a series of convolutional layers.

The subsequent stages of the network adhere to the DPN neural network structure, employing 32 convolutional layers in each sub-block, similar to ResNeXt [35]. The structure of each sub-block follows a bottleneck style [29], comprising a $1 \times 1$ convolutional layer followed by a $3 \times 3$ convolutional layer, and concluding with another $1 \times 1$ convolutional layer. The final $1 \times 1$ convolutional layer is divided into two paths, one connecting the residual path and the other linking the densely connected path.

## Experiments

In this section, we present the details of the datasets used in the development and experimental verification of the proposed AMD-FV. This is followed by a brief overview of the system implementation.

### Datasets

Extensive experiments on five well-known FV benchmarks were conducted, including Labeled Faces in the Wild (LFW) [36], Megaface [37], IARPA Janus Benchmark B (IJB-B)

**Table 1. Network architecture of the proposed Dual Path Network+.** The parameter $k$ (where $k = 1, 2, 3, ...$) denotes the width increment on the densely connected path, while $G$ represents the cardinality of the groups.

| Stages | DPN+ |
|---|---|
| Block-1 | $3 \times 3, 64, stride = 1$<br>$BatchNormalization$<br>$Relu(Activation)$<br>$BatchNormalization$<br>$3 \times 3, 64, stride = 1$<br>$BatchNormalization$<br>$Relu(Activation)$<br>$3 \times 3, 64, stride = 1$<br>$BatchNormalization$ |
| Block-2 | $3 \times 3 maxpool, stride = 2$<br>$\begin{bmatrix} 1 \times 1, 128, \\ 3 \times 3, 128, G = 32, \\ 1 \times 1, 256(+16) \end{bmatrix}$ x 3 |
| Block-3 | $\begin{bmatrix} 1 \times 1, 256, \\ 3 \times 3, 256, G = 32, \\ 1 \times 1, 512(+32) \end{bmatrix}$ x 6 |
| Block-4 | $\begin{bmatrix} 1 \times 1, 512, \\ 3 \times 3, 512, G = 32, \\ 1 \times 1, 1024(+32) \end{bmatrix}$ x 10 |
| Block-5 | $\begin{bmatrix} 1 \times 1, 1024, \\ 3 \times 3, 1024, G = 32, \\ 1 \times 1, 2048(+64) \end{bmatrix}$ x 3 |
| FC | $\begin{bmatrix} 1 \times 1, 2048, \end{bmatrix}$ x 2 |

[38], Cross-Age LFW (CALFW) [39], and Cross-Pose LFW (CPLFW) [40]. These datasets encompass diverse conditions such as pose, gender, occlusion, and illumination variations, detailed statistics of the datasets are provided in Table 2.

## Implementation details

This section systematically provides the implementation details, including the neural network training configuration, data preprocessing pipeline, evaluation protocols, and the computational environment. The code is available at https://github.com/zeeshanahmedkhan/AML.

**Training Configuration**: The experiments were conducted on a hardware setup of three NVIDIA TITAN X GPUs interconnected via NVLink, with a global batch size of 128 (distributed as 42-43 samples per GPU). Synchronized batch normalization was used to ensure consistent gradient updates across GPUs. The learning rate began at 0.1 with a 5-epoch linear warmup phase, followed by a cosine annealing schedule that gradually reduced the rate to a minimum of 1e-6. Stochastic Gradient Descent (SGD) with momentum ($\mu = 0.9$) was employed as the optimizer, coupled with gradient clipping to a maximum L2 norm of 5.0 to prevent exploding gradients. The regularization strategy included weight decay (5e-4 applied exclusively to the convolutional and fully connected layers), label smoothing ($\epsilon = 0.1$), and stochastic depth with a DropPath probability of 0.2.

**Data Preprocessing Pipeline**: Face detection was performed using MTCNN [41] with 5-landmark localization (eyes, nose, and mouth corners). Alignment involved a similarity transformation based on eye coordinates, supplemented by 3D face reconstruction for images with extreme poses (yaw or pitch exceeding 45°). Augmentation strategies included random horizontal flipping (50% probability), color jitter (brightness ±0.2, contrast ±0.15), and patch masking (up to 15% of the image area). Input normalization comprised channel conversion, per-channel mean subtraction ([91.4953, 103.8827, 131.0912]), and 8-bit quantization with histogram equalization.

**Evaluation Protocols**: For standard benchmarks such as LFW [36], CALFW [39], and CPLFW [40], we adopted a 10-fold cross-validation protocol with unrestricted settings. MegaFace [37] evaluation followed the "Large" protocol (1 million distractor images), while IJB-B [38] verification metrics (TAR@FAR) were computed over 10,000 genuine and 8 million impostor pairs. All face embeddings were compared using cosine similarity, with decision thresholds optimized via the ROC convex hull method.

**Computational Environment**: The implementation used PyTorch 1.12.1 with CUDA 11.6, leveraging mixed FP16/FP32 precision through NVIDIA Apex AMP. Parallelization was managed via PyTorch's DataParallel module with NCCL backend, and all random processes were fixed to a seed value of 42 for reproducibility purposes.

**Table 2. Benchmark datasets.**

|          | Datasets    | Number of subjects | Number of Images |
|----------|-------------|--------------------|------------------|
| Training | MS1Mv2 [10] | 85K                | 3.8M             |
| Testing  | LFW [36]    | 5.7K               | 13K              |
|          | CALFW [39]  | 5.7K               | 13K              |
|          | CPLFW [40]  | 5.7K               | 13K              |
|          | Megaface [37] | 690K             | 1M               |
|          | IJB-B [38]  | 1.8K               | 21K              |

## Results

In this section, we provide an extensive ablation study, followed by a performance comparison with state-of-the-art models for face verification. The ablation study demonstrates the impact of the proposed DPN+ architecture and AML on system performance across the various datasets. Last, we provide a discussion of the main outcomes of performance evaluation.

The performance metrics used in the subsequent sub-sections are as follows:

**True Positive Rate (*TPR*)**: *TPR* is defined as the ratio of samples identified as true positives (*TP*) over the total number of positives (*P*):

$$TPR = \frac{TP}{P}$$

The total number of samples identified as positive is given by:

$$P = TP + FN$$

where *FN* relates to the false negatives.

**False Positive Rate (*FPR*)**: *FPR* is defined as the ratio of samples identified as false positives (*FP*) over the total number of positives (P):

$$FPR = \frac{FP}{P}$$

**Mean Verification Accuracy (Accuracy):** This metric measures the overall correctness of the model in verifying identities. It is defined as the ratio of correctly predicted instances to the total instances:

$$Accuracy = \frac{TP + TN}{TP + TN + FP + FN}$$

**Precision**: Precision quantifies the accuracy of the positive predictions made by the model. It is calculated as the ratio of true positives to the sum of true positives and false positives:

$$Precision = \frac{TP}{TP + FP}$$

**Recall**: Recall measures the model's ability to identify all relevant instances. It is defined as the ratio of true positives to the sum of true positives and false negatives:

$$Recall = \frac{TP}{TP + FN}$$

**True Acceptance Rate (*TAR*)**: *TAR* is measured as the ratio of the number of positive matches, $N_p$, over the sum of the number of positive matches and the number of false rejects, $N_r$:

$$TAR = \frac{N_p}{N_p + N_r}$$

**False Acceptance Rate (*FAR*)**: A complimentary metric to *TAR*, *FAR* relates to the proportion of false accepts, i.e., *FP*, which are incorrectly accepted as genuine attempts, over the total number of impostor attempts, i.e., the sum of *FP* and *TN*:

$$FAR = \frac{FP}{FP + TN}$$

**Rank-1 score**: This metric provides a measure of the FV system's ability to correctly identify the top match for a given query image from a set of images. It is defined as the proportion of queries where the correct match appears as the top-ranked candidate in the system's output:

$$Rank\text{-}1 = \frac{M}{Q}$$

where $M$ and $Q$ relate to the number of queries where the true match ranks as top, and the total number of queries, respectively.

## Ablation study

The ablation study provides evidence in support of the justification for the key components of AMD-FV. Five models are evaluated, using the architecture of a 68 layered DPN+ neural network. Their summary descriptions are shown in Table 3.

The models were analyzed according to several performance metrics, such as model complexity (i.e., parameter number and model size), computational time measured in FLOPs, inference time in seconds, and accuracy rate. Note that model complexity, computational time, and inference time are closely related to the computational power of the machine, whereas accuracy rate is related to model performance on the input data. Tables 4 and 5 illustrate the computational complexity and inference times of the evaluated models, respectively. It should be noted that the same CNN architecture was used in all of the models. Therefore, the chosen metrics will have similar values, providing the means to perform a fair comparison of impact of loss functions on the basis of the accuracy rate.

## Robustness of DPN+

In this sub-section, we focus on the analysis of the proposed Dual Path Network+ over the Dual Path Network, so as to provide a fair comparison. Both DPN and DPN+ were trained

**Table 3. Overview of experimental models.**

| Model Name | Neural Network | Loss Function |
|---|---|---|
| DPN+68-A | DPN+ | Softmax Loss |
| DPN+68-B | DPN+ | ArcFace Loss |
| DPN+68-C | DPN+ | CosFace Loss |
| DPN+68-D | DPN+ | AML |
| DPN68-E | DPN | CosFace Loss |

**Table 4. Comparison of the proposed models w.r.t. Performance Overhead.**

| Models | Performance Overhead | | |
|---|---|---|---|
| | Model Complexity | | Computational Time (FLOPs) |
| | Parameters | Model Size | |
| DPN+68-A | $35.4 \times 10^6$ | 551.0 MB | $14.0 \times 10^9$ |
| DPN+68-B | $35.4 \times 10^6$ | 550.7 MB | $14.0 \times 10^9$ |
| DPN+68-C | $35.4 \times 10^6$ | 550.7 MB | $14.0 \times 10^9$ |
| DPN+68-D | $35.4 \times 10^6$ | 550.7 MB | $14.0 \times 10^9$ |

**Table 5. Comparison of the proposed models w.r.t. inference time.**

| Datasets | Inference Time (*s*) | | | |
|---|---|---|---|---|
| | DPN+68-A | DPN+68-B | DPN+68-C | DPN+68-D |
| LWF | 122.4 | 122.4 | 122.4 | 122.4 |
| CALFW | 118.5 | 118.5 | 118.5 | 118.5 |
| CPLFW | 118.4 | 118.4 | 118.4 | 118.4 |

on the same dataset (MS1Mv2 [10]), under the same settings (See Section Implementation details for details) and loss function (i.e., CosFace loss [9] ). The models were then evaluated on the five benchmark face datasets. The obtained results support the claim that the proposed DPN+ model (DPN+68-C) extracts more discriminative features than those of the DPN model (DPN68-E). The verification accuracy of DPN+68-C is 0.17%, 1.78% and 2.94% higher than that of DPN68-E on the LFW, CPLFW and Megaface datasets, respectively (Table 9). A slight drop of 0.07% in the performance of the DPN+68-C compared to DPN68-E is observed in the case of the CALFW dataset. The slight drop in performance of the DPN+68-C model in Table 9 on the CALFW dataset occurs because the model has not been specifically trained to handle age gaps in photos of the same person. While it excels at matching faces with varying angles (CPLFW), crowded scenes (MegaFace), and general recognition (LFW), aging variations such as wrinkles, facial shape-shifts are distinct and unique challenges. To address this, state-of-the-art approaches employ dedicated techniques, e.g., synthetically "aged" face examples during training, which the proposed model does not provide for. However, DPN+'s stronger performance on pose variations and complex real-world scenarios suggests it is better suited for practical applications, where camera angles and crowded environments matter more than extreme age differences. To address aging variations within the current framework, it is possible to augment the training set with age-specific instances or perform fine-tuning for aging effects, without compromising on the model's current strengths. Finally, the ROC curves in Fig. 2 demonstrate that the DPN+68-C model significantly surpasses the DPN68-E model on the IJB-B dataset.

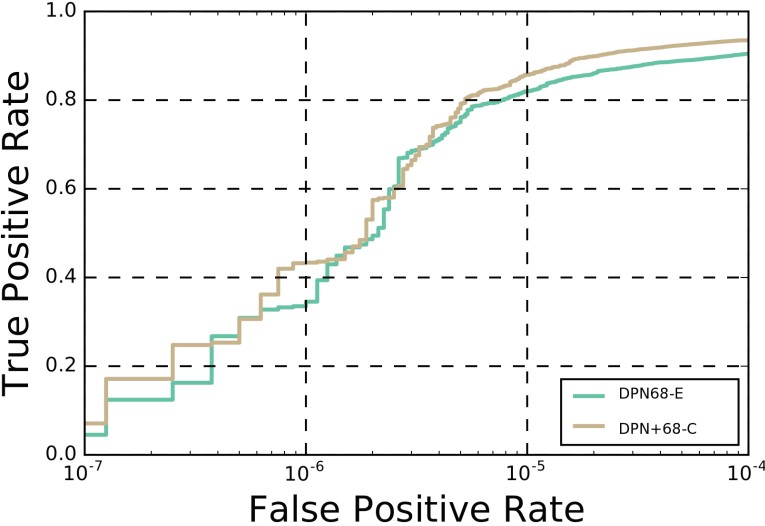

**Fig 2. ROC comparison of DPN68-E and DPN+68-C on the IJB-B dataset.**

## Robustness of AMD-FV

Next, we provide a comparison between the most relevant state-of-the-art angular loss function models, and the proposed AMD-FV model. The SphereFace model [8] proposes a multiplicative angular margin, where the margin value is multiplied by the angle, which, in turn, provides for a fixed distance between the samples. The multiplicative property of the margin highly influences the angle. When the angle decreases towards zero, the margin vanishes (see Fig. 5(a)). The CosFace[9] and ArcFace models [10] introduce the additive angular margin. In CosFace model [9], the margin value is subtracted from the cosine value (Eq. 2), which overcomes the limitation of the vanishing margin (see Fig. 5(b)). Compared with the CosFace model [9], the ArcFace model [10] adds a margin to the angle. The ArcFace model [10] also has a vanishing margin problem. However, because of the additive property, a fixed margin value does not significantly affect the angle (see Fig. 5(c)). In the proposed model, AML automatically estimates the margin using the input data, so that it does not vanish. Specifically, it uses the cross product of the feature map to estimate the margin according to Eq. 3 and subtracts the margin from the cosine value (Eq. 5) (see Fig. 5(d)). By adaptive estimation of the margin and the scale hyperparameters, AML provides an alternative to the otherwise, time-consuming training of the trial and error approach.

The principal objective in this research is the development of an automated procedure for the selection of the margin and scale hyperparameters, which resulted to the proposed AML function. Analysis of the experimental results demonstrates that the accuracy obtained by the AMD-FV model surpasses the performance of both state-of-the-art AdaptiveFace [11] and Adacos models [12]. Note that the mean verification accuracy on the LFW dataset is almost saturated and thus, a small increase in the accuracy has a significant impact. The verification accuracy achieved by the AMD-FV model is 99.75%, which is higher than that of AdaptiveFace at 99.63% and Adacos at 99.71% (Table 6). In addition, the accuracy of the AMD-FV model is higher than that of the baseline model on all the testing datasets. It is worth noting that the introduction of AML results to an average improvement of 2.338% compared to using CESL (Table 10 and Fig. 3).

The verification accuracies obtained by the DPN+68-B model and the DPN+68-D model are similar to one another (Table 8). Specifically, the use of ArcFace loss led to outperforming the DPN+68-D model by a margin of 0.19%, on average, suggesting that the proposed adaptive DPN+68-D model not only surpasses the state-of-the-art adaptive margin-based models, but also favorably compares with the manually optimized margin-based model (i.e., ArcFace). Fig. 4 provides a visual comparison of the computational complexity of the DPN+68-D model and the ArcFace model [10] (i.e., ResNet with 100 layers). It is shown that the proposed AMD-FV model leads to substantial reductions of 32%, 26.92% and 41.91% in terms of the number of layers, parameters and FLOPS, respectively. Moreover, Table 7 compares the performance of two DPN+ architectures, i.e., DPN+68-B (trained with ArcFace loss) and DPN+68-D (trained with AML loss). The results reveal a marginal performance differential between the two models. DPN+68-D exhibits near-equivalent accuracy to DPN+68-B, with

**Table 6. Mean verification accuracy (%) of models, based on the adaptive estimation of the scale and the margin hyperparameters on the LFW Dataset.**

| Models | Accuracy (%) |
| --- | --- |
| AdaptiveFace [11] | 99.62 |
| Adacos [12] | 99.71 |
| AML-FV (DPN+68-D) | **99.75** |

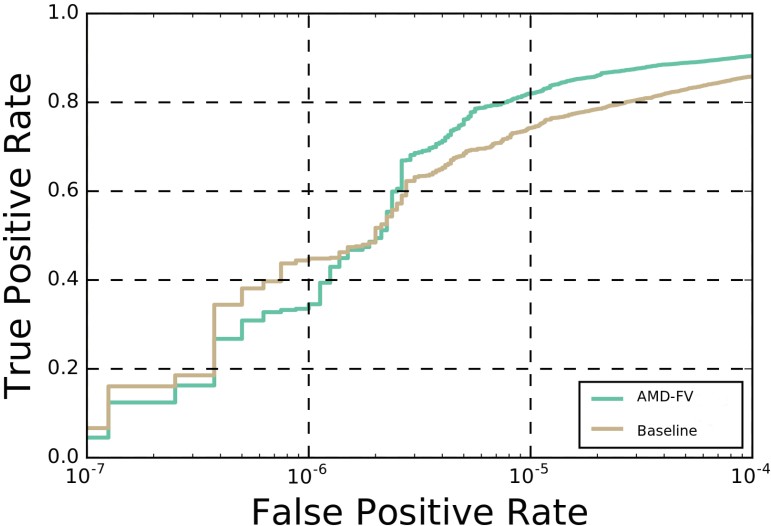

**Fig 3. ROC comparison of AMD-FV and baseline model on the IJB-B dataset.**

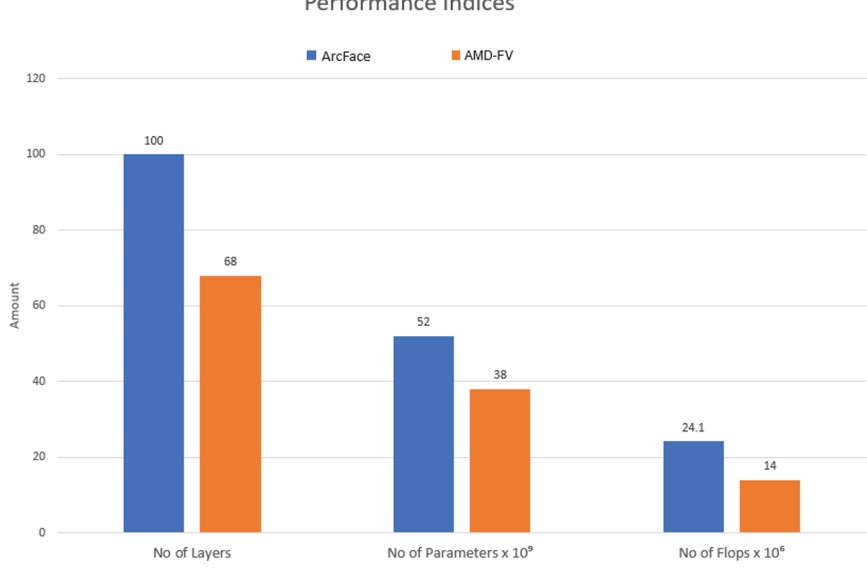

**Fig 4. Comparison of ArcFace and AMD-FV Network Architectures.**

minor deviations, i.e., -0.03% on LFW, -0.19% on CALFW, and -0.35% on CPLFW. Notably, DPN+68-D outperforms DPN+68-B by +0.28% on the IJB-B benchmark.

Tables 7, 8, 11 and 13 demonstrate that ArcFace achieves marginally superior performance on certain benchmarks, however, this apparent advantage stems from ArcFace's reliance on manually optimized hyperparameters, performed on each of the employed datasets. In contrast, AML eliminates the need for manual tuning by automatically adapting the margin and scale parameters based on input dissimilarity and class distribution. This automation inherently introduces trade-offs as follows:

**Table 7. Comparison of mean verification (MV) rates (%) obtained using the ArcFace loss function and the AMD-FV model w.r.t LFW, CALFW, CPLFW and IJB-B Datasets. The best result for each dataset is presented in bold.**

| Models | Testing Datasets | | | |
|---|---|---|---|---|
| | LFW | CALFW | CPLFW | IJB-B |
| | (MV %) | (MV %) | (MV %) | ( TAR@FAR (0.01) %) |
| DPN+68-B | **99.78** | **95.70** | **91.35** | 97.10 |
| DPN+68-D | 99.75 | 95.51 | 91.00 | **97.38** |

**Table 8. Comparison of mean verification accuracy (MVA), precision (Prec) and recall (Rec) in (%) of DPN+68-B model and the DPN+68-D model w.r.t LFW, CALFW and CPLFW Datasets.**

| Models | Testing Datasets | | | | | | | | |
|---|---|---|---|---|---|---|---|---|---|
| | LFW | | | CALFW | | | CPLFW | | |
| | MVA | Prec | Rec | MVA | Prec | Rec | MVA | Prec | Rec |
| DPN+68-B (ArcFace Loss) | **99.78** | **92.267** | **97.563** | **95.70** | 97.603 | **93.815** | **91.35** | **92.249** | **90.357** |
| DPN+68-D (AML-DPN+) | 99.75 | 92.104 | 97.206 | 95.51 | **97.974** | 93.259 | 91.00 | 92.196 | 89.930 |

**Table 9. Comparison of DPN68-E and DPN+68-C of 68 Layers w.r.t verification accuracy (%) on benchmark datasets. Both the models are trained with the use of CosFace loss [9].**

| Datasets | DPN68-E | DPN+68-C |
|---|---|---|
| LFW [36] | 99.61 | **99.78** |
| CALFW [39] | **95.88** | 95.81 |
| CPLFW [40] | 89.63 | **91.41** |
| Megaface [37] | 94.20 | **97.14** |

**Table 10. Comparison of the AMD-FV model with the baseline model w.r.t verification accuracy (%) on benchmark datasets. The best result for each dataset is presented in bold.**

| Datasets | Baseline (DPN+68-A) | AMD-FV (DPN+68-D) |
|---|---|---|
| LFW [36] | 99.68 | **99.75** |
| CALFW [39] | 94.86 | **95.51** |
| CPLFW [40] | 89.16 | **91.00** |
| Megaface [37] | 85.37 | **92.16** |

**Manual Tuning vs. Automation**: ArcFace's manual margin selection allows fine-grained optimization for benchmark metrics, however, it requires extensive trial-and-error, which is impractical for large-scale, real-world deployment. Instead, AML replaces this process with data-driven adaptation, sacrificing marginal accuracy gains for generalizability and scalability.

**Computational Efficiency**: As highlighted in Fig. 5, AML respectively reduces model complexity by 32%, 26.9% and 41.9% in terms of the number of layers, parameters, and FLOPs compared to ArcFace. This efficiency makes AML more suitable for resource-constrained environments.

**Performance Contextualization**: The slight performance gap is negligible in practical applications, however, it underscores the cost of automation. AML avoids the laborious hyperparameter search, making it a preferable choice for dynamic datasets, where manual tuning is infeasible.

**Table 11. Comparison of mean verification accuracy (%) on the LFW dataset. The best result is presented in bold.**

| Models | No. of Models | Training Dataset Size | Acc |
|---|---|---|---|
| Deep-face [45] | 3 | 4M | 97.35 |
| VGG-Face [46] | 1 | 2.6M | 98.95 |
| DeepID2 [44] | 1 | 300K | 98.70 |
| Center Loss [47] | 1 | 0.7M | 99.28 |
| DeepID2 [44] | 25 | 300K | 99.47 |
| RangeLoss [43] | 1 | 1.5M | 99.52 |
| DeepID3 [44] | 25 | 300K | 99.53 |
| FaceNet [42] | 1 | 200M | 99.63 |
| SphereFace [8] | 1 | 0.49M | 99.42 |
| CosFace [9] | 1 | 5M | 99.73 |
| ArcFace [10] | 1 | 5.8M | **99.83** |
| AdaptiveFace [11] | 1 | 5M | 99.62 |
| Adacos [12] | 1 | 6M | 99.71 |
| AMD-FV | 1 | 5.8M | 99.75 |

## Comparison with state-of-the-art methods

In this sub-section, we consider the performance of AMD-FV in regards to related state-of-the-art approaches. The mean verification accuracies acquired on the LFW dataset are presented in Table 11. The proposed AMD-FV method attains a competitive mean verification accuracy of 99.83%, surpassing the performance of the majority of previously published methods e.g., [42], [8], [43], and [44]. Arcface [9] exhibits a slightly higher accuracy compared to AMD-FV by 0.08%. However, in comparison to the manually optimized CosFace model [9], the proposed AMD-FV method with automated hyperparameter selection demonstrates an improvement of 0.02%. This suggests that the automated hyperparameter selection process of AMD-FV could serve as an alternative to the empirically determined hyperparameter values of the CosFace model, eliminating the need for time-consuming trial and error. Furthermore, in comparison to the other two adaptive models, namely AdaptiveFace [11] and Adacos [12], AMD-FV demonstrates a marginal improvement of 0.13% and 0.04%, respectively.

IJB-B is considered to be a challenging dataset concerning occlusion, poses, and illumination. In Table 12, the True Acceptance Rate (TAR) is presented concerning various False Acceptance Rates (FAR) on the IJB-B dataset. The proposed AMD-FV model outperforms

**Table 12. Comparison of 1:1 verification models w.r.t True Acceptance Rate (TAR) at False Acceptance Rates (FAR) of 0.001, 0.01, and 0.1 on the IJB-B dataset. The best results in each column are presented in bold.**

| | 1:1 Verification of TAR | | |
|---|---|---|---|
| | @FAR (0.001) | @FAR (0.01) | @FAR (0.1) |
| VGGFace [5] | 0.711 | 0.850 | – |
| MS1M [5] | 0.857 | 0.935 | – |
| VGGFace2 [5] (ResNet) | 0.878 | 0.938 | 0.975 |
| VGGFace2 [5] (SENet) | 0.888 | 0.949 | 0.984 |
| MN-V [48] | 0.902 | 0.955 | 0.984 |
| MN-VC [48] | 0.909 | 0.958 | 0.985 |
| AMD-FV | **0.948** | **0.973** | **0.989** |

**Table 13. Comparison of Rank-1 identification scores (%) achieved by various methods on the Megaface dataset. The best result is presented in bold.**

|  | Protocol | Rank-1 Identification Score |
|---|---|---|
| Softmax [8] | Small | 54.85 |
| Contrastive Loss [8] | Small | 65.21 |
| Triplet Loss [8] | Small | 64.79 |
| Center Loss [47] | Small | 65.49 |
| SphereFace [8] | Small | 72.72 |
| CosFace [9] | Small | 77.11 |
| CosFace [9] | Large | 82.72 |
| FaceNet [42] | Large | 70.49 |
| ArcFace [10] | Large | **98.35** |
| Adacos [12] | Large | 97.41 |
| AdaptiveFace [11] | Large | 95.02 |
| AMD-FV | Large | 92.16 |

all other models. Specifically, it improves on the VGGFace2 model by 6%, and by 4.6% and 3.9% compared to MN-V and MN-VC, respectively, at a FAR of 0.001. At a FAR of 0.01, the proposed AMD-FV surpasses VGGFace2, MN-V, and MN-VC models by 2.83%, 2.23%, and 1.93%, respectively. At a FAR of 0.1, AMD-FV's performance marginally outperforms that of the VGGFace2, MN-V, and MN-VC. This insight underscores the improved performance of AMD-FV compared to state-of-the-art methods on the IJB-B dataset.

The Rank-1 identification scores of existing methods on the Megaface dataset are presented in Table 13. The gallery set of 1M distractor images in Megaface presents a wide range of poses, ages, and illumination changes, making this dataset exceptionally challenging for face verification. Notably, the proposed AMD-FV method outperforms several state-of-the-art models, such as the CosFace model [9], Sphereface model [8], and FaceNet [42], with considerable margins (i.e., greater than 10%). However, when compared to the adaptive hyper-parameter estimation models of AdaptiveFace [11] and Adacos [12], the results of AMD-FV are lower. This difference is attributed to Megaface's use of a large number of distractor images in its verification protocol, which affects AMD-FV's margin hyperparameter estimation values computed based on the mean. It is worth noting that the primary objective of this work is to adaptively estimate the large margin cosine loss of CosFace [9]. In this regard, the proposed AMD-FV model achieves a Rank-1 score of 92.16%, significantly surpassing CosFace by 9.44%.

The mean verification accuracy on the Cross Age LFW (CALFW) [39] and Cross Pose LFW (CPLFW) datasets [40] is presented in Table 14. These datasets extend the LFW dataset by introducing variations in age and pose, thereby increasing the complexity level. On the CALFW dataset, AMD-FV significantly outperforms the Center loss [47], SphereFace [8], and VGGFace2 models [5] by 10%, 5.21%, and 4.94%, respectively. Similarly, on the CPLFW

**Table 14. Comparison of mean verification accuracy (%) on the CALFW and CPLFW datasets [39] [40]. The best result is presented in bold.**

|  | CALFW | CPLFW |
|---|---|---|
| Center Loss [47] | 85.48 | 77.48 |
| SphereFace [8] | 90.30 | 81.40 |
| VGGFace2 [5] | 90.57 | 84.00 |
| AMD-FV | **95.51** | **91.00** |

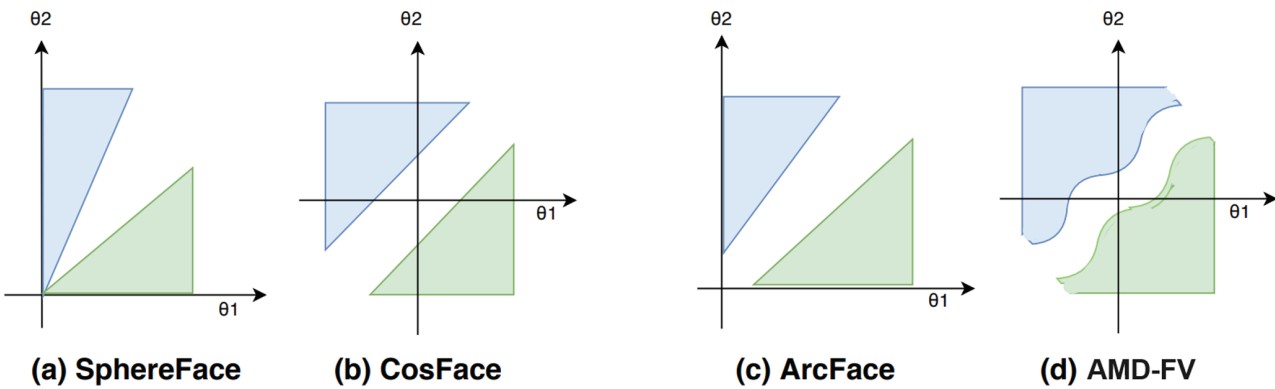

**Fig 5. Geometric representation of angular loss functions.** (a) SphereFace [8], (b) CosFace [9], (c) ArcFace [10], and (d) AMD-FV. Please note that the setting of binary classification is considered.

dataset, the mean verification accuracy of AMD-FV surpasses that of the Center loss [47], SphereFace [8], and VGGFace2 models [5] by 13.52%, 9.60%, and 7.00%, respectively.

## Discussion

A fair comparison was conducted to evaluate the performance of the proposed Dual Path Network+ over the Dual Path Network. Both DPN and DPN+ were trained using the same dataset (MS1Mv2 [10]), under identical settings (see Implementation details for more details).

The proposed Adaptive Margin Loss automatically estimates the margin using the input data. Specifically, it utilizes the cross product of the feature map to estimate the margin according to Eq 3 and then subtracts the margin from the cosine value (Eq 5) (see Fig 5(d)). Through adaptive estimation of the margin and scale hyperparameters, AML offers an alternative to the conventional time-consuming trial-and-error training approach.

The overall results demonstrate that the proposed AMD-FV model outperforms the majority of the state-of-the-art methods. This advancement stems from three primary contributions:

**DPN+ Contribution:** Table 9 demonstrates that DPN+68-C outperforms DPN68-E by 0.17% on LFW and 1.78% on CPLFW. This improvement stems directly from DPN+'s redesigned Block-1, which preserves low-level spatial features through three sequential 3x3 convolutions (vs. DPN's 3x3 maxpool with stride=2). The absence of downsampling in Block-1 reduces information loss, particularly benefiting pose-sensitive datasets like CPLFW.

**AML Contribution:** Table 7 reveals that AML in DPN+68-D model achieves comparable accuracy to ArcFace loss in DPN+68-B model despite the use of automated hyperparameter selection (i.e., 99.75% vs. 99.78%) on LFW. This marginal drop of 0.03% reflects the cost of replacing manual tuning with AML's data-driven adaptation. However, AML reduces training complexity by eliminating tedious and laborious effort on manual hyperparameter search.

**Integrating DPN+ with AML:** The combined AMD-FV model achieves 41.9% lower FLOPs than ArcFace (Fig. 4) by combining AML's efficient scaling with DPN+'s parameter reduction.

## Conclusions

This work presents AMD-FV, an enhanced deep-learning model for face verification, which capitalizes on two key improvements. The first contribution addresses the issue of information

loss caused by downsampling inherent in the Dual Path Network. We introduce the Dual Path Network+, incorporating a novel Block-1 with 3x3 convolutions, batch normalization, and the ReLU activation function. DPN+ offers a favorable alternative in terms of computational cost, including fewer layers, fewer parameters, and lower FLOPS than ResNet. The second contribution addresses the tedious and time-consuming process of trial and error when selecting the margin and scale hyperparameters of angular loss functions. We propose the Adaptive Margin Loss function, which automates hyperparameter selection, eliminating the need for extensive experimentation typically required by state-of-the-art methods. Overall, the AMD-FV model demonstrates competitive performance with the advantages of automated hyperparameter selection. Specifically, the proposed framework achieves verification accuracy of 99.75% on the LFW dataset, outperforming AdaptiveFace (99.62%) and Adacos (99.71%). Moreover, it delivers TAR of 94.8%, 97.3% and 98.9% at FAR of 0.001, 0.01 and 0.1, respectively, on the IJB-B dataset, surpassing VGGFace2 by up to 6%. Similarly, competitive performances are delivered on Megaface with Rank-1 identification scores of 92.16%, compared to CosFace, and MVA of 95.51% and 91%, respectively, on CALFW and CPLFW. In regards to computational efficiency, AMD-FV achieves significant reductions in terms of the number of layers (32%), system parameters (26.92%) and FLOPs (41.91%), compared to ArcFace, positioning it as an attractive solution to real-world applications. A number of potential directions will be pursued as part of further research, including cross-domain applications, e.g., expand the use of AMD-FV in person re-identification, iris recognition, and fingerprint recognition, thus evaluating its adaptability. Moreover, we intend to combine the proposed system with other biometric modalities, for instance, gait, and consider its authentication accuracy in complex scenarios. Further research will evaluate its resilience against adversarial attacks to ensure security in such critical applications.

## Supporting information

**S1 Appendix:** Magnitude of Cross Product
(PDF)

## Author contributions

**Conceptualization:** Zeeshan Ahmed Khan, Panos Liatsis.

**Data curation:** Zeeshan Ahmed Khan.

**Investigation:** Zeeshan Ahmed Khan.

**Methodology:** Zeeshan Ahmed Khan, Waqar Ahmed, Panos Liatsis.

**Project administration:** Panos Liatsis.

**Software:** Zeeshan Ahmed Khan, Waqar Ahmed.

**Supervision:** Panos Liatsis.

**Validation:** Zeeshan Ahmed Khan, Panos Liatsis.

**Visualization:** Zeeshan Ahmed Khan, Waqar Ahmed.

**Writing – original draft:** Zeeshan Ahmed Khan, Panos Liatsis.

**Writing – review & editing:** Waqar Ahmed, Panos Liatsis.

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
