## [Decision Letter · Decision Letter 0]

4 Nov 2024

PONE-D-24-28461AMD-FV: Adaptive Margin Loss and Dual Path Network+ for Deep Face VerificationPLOS ONE

Dear Dr. Liatsis,

Thank you for submitting your manuscript to PLOS ONE. After careful consideration, we feel that it has merit but does not fully meet PLOS ONE’s publication criteria as it currently stands. Therefore, we invite you to submit a revised version of the manuscript that addresses the points raised during the review process.

We look forward to receiving your revised manuscript.

Kind regards,

Lei Chu

Academic Editor

PLOS ONE

3. Thank you for uploading your study's underlying data set. Unfortunately, the repository you have noted in your Data Availability statement does not qualify as an acceptable data repository according to PLOS's standards. At this time, please upload the minimal data set necessary to replicate your study's findings to a stable, public repository (such as figshare or Dryad) and provide us with the relevant URLs, DOIs, or accession numbers that may be used to access these data. For a list of recommended repositories and additional information on PLOS standards for data deposition, please see https://journals.plos.org/plosone/s/recommended-repositories.

Additional Editor Comments:

We have gathered comments from two reviewers, and we believe their feedback is constructive and valuable. Please revise the manuscript accordingly.

Reviewers' comments:

Reviewer's Responses to Questions

**Comments to the Author**

1. Is the manuscript technically sound, and do the data support the conclusions?

Reviewer #1: Yes

Reviewer #2: Partly

2. Has the statistical analysis been performed appropriately and rigorously? 

Reviewer #1: Yes

Reviewer #2: No

3. Have the authors made all data underlying the findings in their manuscript fully available?

Reviewer #1: Yes

Reviewer #2: Yes

4. Is the manuscript presented in an intelligible fashion and written in standard English?

Reviewer #1: Yes

Reviewer #2: Yes

5. Review Comments to the Author

Reviewer #1: 1- The abstract lacks an explanation of the proposed algorithm in simple and clear steps, in addition to not stating the type of metrics and their values used to demonstrate the efficiency of the proposed algorithm.

2- No keywords found

3- The abbreviation is added to the scientific name only at the first appearance and not at every appearance, for example Adaptive Margin Loss (AML).

4- Ensure that the research papers included in the Related Works section are arranged according to the year of publication (from oldest to newest). In addition to stating the negatives found in all or some of these papers.

5- The comparison presented by the researcher between the submitted work and other works needs more clarification and expansion in the discussion of the included tables and charts.

6-No numerical values were included for the results in the conclusions addressed by the researcher.

7- There are no published references in 2024, so it is preferable to include at least two research papers published in the above-mentioned year.

Reviewer #2: This manuscript presents a Face verification based on Adaptive Margin Loss and Dual Path Network+. Although there are extensive experiments, the author did not clarify the appropriate motivations and limitations. Please address the following comments for your next revisions.

1- The fourth contribution is not really a contribution. This is a standard procedure for verification of the performance of the proposed method.

2- The motivation seems weak, please try to focus on other methods limitations and try to optimize your work accordingly.

3- The related work section is limited to few works, and there are no expansion to other pipelines, this should be reflected on your work motivations. Try to reorganize your related work section to mathematical models, ML and DL models.

4- There is no prof of concept in each step in your work.

5- Where is the ablation study. Add this subsection to your experimental results sections.

6- There are no mention of the metrics equations, and how did you verify your results with these metrics.

6. PLOS authors have the option to publish the peer review history of their article (what does this mean?). If published, this will include your full peer review and any attached files.

Reviewer #1: No

Reviewer #2: **Yes: **Firas Abedi

---

## [Author Response · Author response to Decision Letter 1]

20 Dec 2024

Reviewer #1:

We would like to thank the reviewer for their rigorous review of our work and their constructive comments, which have led to substantial improvement in the presentation and communication of the theory and results presented in this work. In what follows

R1.C1- The abstract lacks an explanation of the proposed algorithm in simple and clear steps, in addition to not stating the type of metrics and their values used to demonstrate the efficiency of the proposed algorithm.

Response: Thank you for your helpful comment. The abstract has been re-written to provide a clear explanation of the proposed approach, including the results of the metrics used to demonstrate performance:

“Face verification is important in a variety of applications, for instance, access control, surveillance, and identification. Existing methods often struggle with the challenges of dataset imbalance and manual tuning of hyperparameters. To address this, we propose the Adaptive Margin Loss and Dual Path Network+ (AMD-FV) for deep face verification. Two innovations are introduced, namely, Adaptive Margin Loss (AML) and Dual Path Network+ (DPN+). AML aims at automating the selection of margin and scale hyperparameters in large margin loss functions, thus, eliminating the need for manual tuning. Input dissimilarity information is used to estimate the margin, while the scale parameter is computed using the number of classes and AML's range. Next, DPN+ enhances the original Dual Path Network by redesigning the first block with a series of 3x3 convolutions, batch normalization, and ReLU activations, leveraging shared connections across layers, leading to increases in spatial resolution and computational cost efficiency, while maximizing the use of discriminative features. We present comprehensive experiments on five diverse face verification datasets (LFW, Megaface, IJB-B, CALFW, and CPLFW) to demonstrate the effectiveness of the proposed approach. The results show that AMD-FV outperforms state-of-the-art methods, achieving a verification accuracy of 99.75% on LFW, improving the True Acceptance Rate by 6% on IJB-B at a False Acceptance Rate of 0.001, compared to VGGFace2, and attaining a Rank-1 identification score of 92.16% on Megaface, surpassing the CosFace model by 9.44%.”

R1.C2- No keywords found.

Response: Thank you for your suggestion. The PLOS One journal template does not explicitly incorporate keywords section, which is the reason why keywords were not identified in the original submission. To accommodate the reviewer’s comment, a new section “Keywords” was added, immediately after the abstract, to capture the key technical terms involved in the proposed work:

“Keywords

Face Verification, Adaptive Margin loss, Dual Path Network+, adaptive hyperparameter selection"

R1.C3- The abbreviation is added to the scientific name only at the first appearance and not at every appearance, for example Adaptive Margin Loss (AML).

Response: We apologize for the oversight. The manuscript has been thoroughly revised to ensure that abbreviations are only defined at first appearance.

R1.C4- Ensure that the research papers included in the Related Works section are arranged according to the year of publication (from oldest to newest). In addition to stating the negatives found in all or some of these papers.

Response: Thank you for the helpful suggestion. References have been arranged in chronological order, and a detailed critique of the works has been provided in the revised manuscript:

“Margin-based loss functions, such as SphereFace [8], CosFace [9], and ArcFace [10], demonstrated significant success in face verification tasks. The hyperparameters associated with these loss functions, often manually chosen, play a crucial role on the performance of the deep learning model. Indeed, suboptimal or incorrect selection of these hyperparameters can significantly affect their properties and, subsequently, model performance. Typically, determining these parameters involves a time-consuming and laborious trial-and-error process, the success of which is limited by the choice of initial parameters. To address this issue, researchers have proposed data-driven methods to estimate these hyperparameters. For instance, Liu et al., [11] argue that face data typically consists of large-scale classes and samples, emphasizing that selecting appropriate classes and sample numbers for training helps the model achieve better generalization. They proposed the use of hard prototype mining and introduced the AdaptiveFace model to estimate the associated hyperparameters. However, the computational overhead of learning individual margins for each class thoroughly considered. In the case of large-scale datasets, this could potentially lead to significant increases in training time and memory requirements. Moreover, as demonstrated in the experiments, the performance of the method is sensitive to the choice of the hyperparameter controlling the strength of the margin constraint. Zhang et al., [12] propose the Adacos model to tackle the challenge of hyperparameter selection in face verification tasks. However, their approach presents a number of challenges. First, it assumed that during training, the sum of logits for non-corresponding classes remains almost constant. Beyond empirical evidence, the theoretical justification for this assumption, crucial in deriving the adaptive scale parameter, is not proven. Moreover, the analysis of probability curves is based on simplified models, which may not fully capture the complexity of real-world face recognition scenarios, e.g., class imbalance and samples of varying difficulty.

Zhang et al., [25], propose diversity regularization by training the margin hyperparameter and constraining it within the range of 0.5 and �. While the authors automatically train the margin hyperparameter, they introduce two additional manual parameters in terms of the range. Moreover, there is insufficient analysis of the impact of the balance hyperparameter across different datasets and network architectures, and the computational overhead associated with the calculation of adaptive margins and diversity regularization. RamFace [26] addresses racial bias by proposing an adaptive margin loss based on intra-identity compactness using multitask learning. Two loss functions, i.e., race adaptive margin-based loss and race classification loss, are introduced, however, the additional hyperparameter � is used to balance them. There is insufficient consideration on the appropriate choice of this hyperparameter, and its impact on model performance. In addition, the manuscript neither provides formal guarantees on the fairness properties of the proposed method, nor does it discuss in sufficient detail the potential trade-offs between overall accuracy and fairness across different racial groups. EnhanceFace [27], proposes adaptive weighted softmax loss by segregating hard samples into semi-hard and harder samples. The loss function adjusts the corresponding weights during training. The authors introduce the enhance rate ω, used to adjust the effects of semi-hard samples. However, the selection of this parameter appears arbitrary, as there is lack of justification for the chosen range and the step size. JAMsFace [28] proposes a joint adaptive margin loss function, which learns class-related margins by incorporating margin penalties within the cosine angle of a loss function. These models mitigate the problem to some extent. However, they only provide a means of estimating a single hyperparameter, i.e., margin, and moreover, there is a lack of a clear formulation on how this is calculated based on class distribution. In contrast, the proposed AML approach estimates both the scale and margin hyperparameters.

In the context of CNN architectures, García et al., [21] discussed the difficulty of preserving diagnostic features in medical images during downsampling. He et al., [29] introduced the Residual Network (ResNet), which utilizes skip connections to enable very deep networks, addressing optimization challenges. However, the use of downsampling in ResNet leads to significant information loss. Sedlar et al., [22] identified similar challenges in genomic signal processing, calling for strategies that balance data reduction with feature retention. Zhao et al., [23] proposed random shifting to minimize systematic information loss during downsampling. In contrast, Huang et al., [30] introduced the Dense Convolutional Network (DenseNet), which uses skip connections to concatenate features from earlier layers, but at the expense of high computational complexity. Chen et al., [20] demonstrated that ResNet reuses features via residual paths, while DenseNet explores new features through densely connected paths. Building on this, Chen et al., [20] proposed the Dual Path Network (DPN), which combines the strengths of both ResNet and DenseNet, although it still suffers from information loss due to its downsampling approach. The proposed Dual Path Network+ is an enhancement to the original DPN, improving spatial resolution and computational efficiency, while leveraging shared connections across layers to maximize feature utilization.”

R1.C5- The comparison presented by the researcher between the submitted work and other works needs more clarification and expansion in the discussion of the included tables and charts.

Response: We are sincerely grateful to the reviewer for their recommendation. To address this comment, we substantially expanded the Results section, by including a comprehensive ablation study section to provide a detailed and comprehensive discussion of the results. This section highlights the improvements introduced by each of the components in the proposed AMD-FV framework, with a clear explanation of how the performance metrics compare to those in state-of-the-art works, as follows:

“Ablation study

The ablation study provides evidence in support of the justification for the key components of AMD-FV. Five models are evaluated, using the architecture of a 68 layered DPN+ neural network. Their summary descriptions are shown in Table 3.

The models were analyzed according to several performance metrics, such as model complexity (i.e., parameter number and model size), computational time measured in FLOPs, inference time in seconds, and accuracy rate. Note that model complexity, computational time, and inference time are closely related to the computational power of the machine, whereas accuracy rate is related to model performance on the input data. Tables 4 and 5 illustrate the computational complexity and inference times of the evaluated models, respectively. It should be noted that the same CNN architecture was used in all of the models. Therefore, the chosen metrics will have similar values, providing the means to perform a fair comparison of impact of loss functions on the basis of the accuracy rate.

Robustness of DPN+

In this sub-section, we focus on the analysis of the proposed Dual Path Network+ over the Dual Path Network, so as to provide a fair comparison. Both DPN and DPN+ were trained on the same dataset (MS1Mv2 [10]), under the same settings (See Section for details) and loss function (i.e., CosFace loss [9] ). The models were then evaluated on the five benchmark face datasets. The obtained results support the claim that the proposed DPN+ model (DPN+68-C) extracts more discriminative features than those of the DPN model (DPN68-C). The verification accuracy of DPN+68-D is 0.17%, 1.78% and 2.94% higher than that of DPN68-C on the LFW, CPLFW and Megaface datasets, respectively (Table 9). A slight drop of 0.07% in the performance of the DPN+68-C compared to DPN68-C is observed in the case of the CALFW dataset. Finally, the ROC curves in Fig. 3 demonstrate that the DPN+68-C model significantly surpasses the DPN68-C model on the IJB-B dataset.

Robustness of AMD-FV

Next, we provide a comparison between the most relevant state-of-the-art angular loss function models, and the proposed AMD-FV model. The SphereFace model [8] proposes a multiplicative angular margin, where the margin value is multiplied by the angle, which, in turn, provides for a fixed distance between the samples. The multiplicative property of the margin highly influences the angle. When the angle decreases towards zero, the margin vanishes (see Fig. 6(a)). The CosFace [9] and ArcFace models [10] introduce the additive angular margin. In CosFace model [9], the margin value is subtracted from the cosine value (Eq. 2), which overcomes the limitation of the vanishing margin (see Fig. 6(b)). Compared with the CosFace model [9], the ArcFace model [10] adds a margin to the angle. The ArcFace model [10] also has a vanishing margin problem. However, because of the additive property, a fixed margin value does not significantly affect the angle (see Fig. 6(c)). In the proposed model, AML automatically estimates the margin using the input data, so that it does not vanish. Specifically, it uses the cross product of the feature map to estimate the margin according to Eq. 3 and subtracts the margin from the cosine value (Eq. 5) (see Fig. 6(d)). By adaptive estimation of the margin and the scale hyperparameters, AML provides an alternative to the otherwise, time-consuming training of the trial and error approach.

The principal objective in this research is the development of an automated procedure for the selection of the margin and scale hyperparameters, which resulted to the proposed AML function. Analysis of the experimental results demonstrates that the accuracy obtained by the AMD-FV model surpasses the performance of both state-of-the-art AdaptiveFace [11] and Adacos models [12]. Note that the mean verification accuracy on the LFW dataset is almost saturated and thus, a small increase in the accuracy has a significant impact. The verification accuracy achieved by the AMD-FV model is 99.75%, which is higher than that of AdaptiveFace at 99.63% and Adacos at 99.71% (Table 6). In addition, the accuracy of the AMD-FV model is higher than that of the baseline model (DPN+68-A) on all the testing datasets. It is worth noting that the introduction of AML results to an average improvement of 2.338% compared to using CESL (Table 10 and Fig. 4). The verification accuracies obtained by the DPN+68-B model (DPN+ with ArcFace loss) and the DPN+68-D model are similar to one another (Table 8). Specifically, the use of ArcFace loss led to outperforing the DPN+68-D model by a margin of 0.19%, on average, suggesting that the proposed adaptive DPN+68-D model not only surpasses the state-of-the-art adaptive margin-based models, but also favorably compares with the manually optimized margin-based model (i.e., ArcFace). Fig. 5 provides a visual comparison of the computational complexity of the DPN+68-D model and the ArcFace model [10] (i.e., ResNet with 100 layers). It is shown that the proposed AMD-FV model leads to substantial reductions of 32%, 26.92% and 41.91% in terms of the number of layers, parameters and FLOPS, respectively.

Comparison with state-of-the-art methods

In this sub-section, we consider the performance of AMD-FV in regards to related state-of-the-art approaches. The mean verification accuracies acquired on the LFW dataset are presented in Table 11. The proposed AMD-FV method attains a competitive mean verification accuracy of 99.83%, surpassing the performance of the majority of previously published methods e.g., [42], [8], [43], and [44]. Arcface [9] exhibits a slightly higher accuracy compared to AMD-FV by 0.08%. However, in comparison to the manually optimized CosFace model [9], the proposed AMD-FV method with automated hyperparameter selection demonstrates an improvement of 0.02%. This suggests that the automated hyperparameter selection process of AMD-FV could serve as an alternative to the empirically determined hyperparameter values of the CosFace model, eliminating the need for time-consuming trial and error. Furthermore, in comparison to the other two adaptive models, namely AdaptiveFace [11] and Adacos [12], AMD-FV demonstrates a marginal improvement of 0.13% and 0.04%, respectively.

IJB-B is considered to be a challenging dataset concerning occlusion, poses, and illumination. In Table 12, the True Acceptance Rate (TAR)

---

## [Decision Letter · Decision Letter 1]

28 Jan 2025

PONE-D-24-28461R1AMD-FV: Adaptive Margin Loss and Dual Path Network+ for Deep Face VerificationPLOS ONE

Dear Dr. Liatsis,

Thank you for submitting your manuscript to PLOS ONE. After careful consideration, we feel that it has merit but does not fully meet PLOS ONE’s publication criteria as it currently stands. Therefore, we invite you to submit a revised version of the manuscript that addresses the points raised during the review process.

Based on the reviewers' comments, while your manuscript presents an interesting approach, significant revisions are required to improve its clarity, reproducibility, and depth of analysis. Key areas for improvement include providing detailed descriptions of the experimental setup, clearly quantifying the contributions of core components to performance gains, elaborating on hyperparameter selection strategies, and ensuring the methodology is sufficiently detailed for reproducibility. We encourage you to address these points comprehensively and resubmit your revised manuscript for further evaluation. 

We look forward to receiving your revised manuscript.

Kind regards,

Lei Chu

Academic Editor

PLOS ONE

Reviewers' comments:

Reviewer's Responses to Questions

**Comments to the Author**

1. If the authors have adequately addressed your comments raised in a previous round of review and you feel that this manuscript is now acceptable for publication, you may indicate that here to bypass the “Comments to the Author” section, enter your conflict of interest statement in the “Confidential to Editor” section, and submit your "Accept" recommendation.

Reviewer #1: All comments have been addressed

Reviewer #3: All comments have been addressed

Reviewer #4: (No Response)

2. Is the manuscript technically sound, and do the data support the conclusions?

Reviewer #1: (No Response)

Reviewer #3: Yes

Reviewer #4: Partly

3. Has the statistical analysis been performed appropriately and rigorously? 

Reviewer #1: (No Response)

Reviewer #3: Yes

Reviewer #4: Yes

4. Have the authors made all data underlying the findings in their manuscript fully available?

Reviewer #1: (No Response)

Reviewer #3: Yes

Reviewer #4: Yes

5. Is the manuscript presented in an intelligible fashion and written in standard English?

Reviewer #1: (No Response)

Reviewer #3: (No Response)

Reviewer #4: Yes

6. Review Comments to the Author

Reviewer #1: (No Response)

Reviewer #3: Thank you for addressing the reviewers' comments. The manuscript has been improved significantly. However, I have some minor comments that need to be addressed.

Comments:

1. No need to recite the models every time they are mentioned.

2. It is unclear how Algorithm 1 obtains the values of N and R used in Eq. (4). Please add these variables to the algorithm's requirements.

3. The name of the DPN model with CosFace is listed as DPN68-E in Table 3 and DPN68-C in Table 9. Please ensure consistency.

4. Please provide one or two sentences to justify the performance drop of DPN+68-C on the CALFW dataset.

5. The results of Table 7 are not discussed in the manuscript.

6. It appears that the ArcFace loss function outperforms the proposed AML in the results shown in Tables 7, 8, 11, and 13. Although there is a brief mention at the end of the “Robustness of AMD-FV” section stating that AML is less complex, I believe further discussion is needed at the beginning of this section.

7. ArcFace has the best result in Table 13, not Adacos.

8. Closing parentheses is missing in table 5. "Inference Time (s)"

9. It should be Fig. 5 instead of Fig. 6 in the last paragraph of the "Discussion" section. The sentence should read: "Fig. 5 provides a visual comparison ..."

Reviewer #4: 1. the experimental setup section could be more detailed, including specific training parameters, optimiser settings, data preprocessing steps, etc.

2. despite the ablation study, the specific contribution of each of the AML and DPN+ components to the performance improvement of the final results is not detailed.

3. although a method for automatic selection of hyperparameters is proposed, the specific range of values and tuning strategies for these hyperparameters are not detailed.

4. The methodology section needs to detail the specific implementation and optimisation strategy of each component to ensure that the proposed methodology can be clearly understood and reproduced by the reader.

7. PLOS authors have the option to publish the peer review history of their article (what does this mean?). If published, this will include your full peer review and any attached files.

Reviewer #1: No

Reviewer #3: No

Reviewer #4: No

---

## [Author Response · Author response to Decision Letter 2]

13 Mar 2025

Reviewer #1:

(No Response)

Reviewer #3:

We would like to thank the reviewer for the rigorous review of our work and their constructive comments, which have led to substantial improvement in the presentation and communication of the theory and results presented in this work. In what follows we provide a point-by-point response to all comments raised by the reviewer.

R3.C1 - No need to recite the models every time they are mentioned.

Response: Thank you for your valuable feedback. We revised the manuscript to eliminate redundant model recitations by consistently using abbreviated notations. These edits streamline the narrative and improve readability.

R3.C2 - It is unclear how Algorithm 1 obtains the values of N and R used in Eq. (4). Please add these variables to the algorithm's requirements.

Response: Thank you for highlighting this ambiguity. We updated Algorithm 1 to explicitly define N (number of training classes) and R (maximum adaptive range of AML, derived from Appendix B) as inputs to the scale hyperparameter computation in Step 5. These variables are now clearly stated in the algorithm description.

R3.C3 - The name of the DPN model with CosFace is listed as DPN68-E in Table 3 and DPN68-C in Table 9. Please ensure consistency.

Response: Thank you for identifying this inconsistency. We rectified this error by updating the model name in Table 9 from "DPN68-C" to "DPN68-E" to align with the nomenclature in Table 3. The revised manuscript now uniformly references the DPN with CosFace loss as "DPN68-E" across all sections, tables, and analyses. We also performed a thorough cross-check to ensure such discrepancies do not persist elsewhere in the document.

R3.C4 - Please provide one or two sentences to justify the performance drop of DPN+68-C on the CALFW dataset.

Response: Thank you for raising this important point. The marginal performance drop of DPN+68-C on CALFW stems from its training focus on pose invariance and real-world robustness, rather than explicit age-invariant modeling. While current leading methods may employ synthetic aging augmentation, our framework prioritizes practical deployment scenarios where pose variations and occlusions are more prevalent than extreme age gaps. We explicitly addressed this in the revised "Robustness of DPN+" section, clarifying its implications and future mitigation strategies, as follows:

“The slight drop in performance of the DPN+68-C model in Table 9 on the CALFW dataset occurs because the model has not been specifically trained to handle age gaps in photos of the same person. While it excels at matching faces with varying angles (CPLFW), crowded scenes (MegaFace), and general recognition (LFW), aging variations, such as wrinkles, facial shape-shifts are distinct and unique challenges. To address this, state-of-the-art approaches employ dedicated techniques, e.g., synthetically "aged" face examples during training, which the proposed model does not provide for. However, DPN+’s stronger performance on pose variations and complex real-world scenarios suggests it is better suited for practical applications, where camera angles and crowded environments matter more than extreme age differences. To address aging variations within the current framework, it is possible to augment the training dataset with aging-specific instances or perform fine-tuning for aging effects, without compromising on the model’s current strengths.”

R3.C5 - The results of Table 7 are not discussed in the manuscript.

Response: Thank you for highlighting this oversight. We revised the “Robustness of AMD-FV” section to include a detailed analysis of Table 7, which compares the performance of DPN+68-B and DPN+68-D models as follows:

“Moreover, Table 7 compares the performance of two DPN+ architectures, i.e., DPN+68-B (trained with ArcFace loss) and DPN+68-D (trained with AML loss). The results reveal a marginal performance differential between the two models. DPN+68-D exhibits near-equivalent accuracy to DPN+68-B, with minor deviations, i.e., -0.03% on LFW, -0.19% on CALFW, and -0.35% on CPLFW. Notably, DPN+68-D outperforms DPN+68-B by +0.28% on the IJB-B benchmark.”

R3.C6 - It appears that the ArcFace loss function outperforms the proposed AML in the results shown in Tables 7, 8, 11, and 13. Although there is a brief mention at the end of the “Robustness of AMD-FV” section stating that AML is less complex, I believe further discussion is needed at the beginning of this section.

Response: We would like to thank the reviewer for raising this comment, which provides us with the opportunity to clarify the comparison between ArcFace loss and AMD-FV. We agree with the reviewer that Tables 7, 8, 11, and 13 demonstrate that ArcFace achieves marginally superior performance on certain benchmarks, however, this apparent advantage stems from ArcFace’s reliance on manually optimized hyperparameters, performed on each of the employed datasets. In contrast, AML eliminates the need for manual tuning by automatically adapting the margin and scale parameters based on input dissimilarity and class distribution. This automation inherently introduces a trade-off as follows:

Manual Tuning vs. Automation: ArcFace’s manual margin selection allows fine-grained optimization for benchmark metrics, however, it requires extensive trial-and-error, which is impractical for large-scale, real-world deployment. Instead, AML replaces this process with data-driven adaptation, sacrificing marginal accuracy gains for generalizability and scalability.

Computational Efficiency: As highlighted in Fig. 5, AML respectively reduces model complexity by 32%, 26.9% and 41.9% in terms of the number of layers, parameters, and FLOPs compared to ArcFace. This efficiency makes AML more suitable for resource-constrained environments.

Performance Contextualization: The slight performance gap is negligible in practical applications, however, it underscores the cost of automation. AML avoids the laborious hyperparameter search, making it a preferable choice for dynamic datasets, where manual tuning is infeasible.

To address the reviewer’s comment, the following discussion has been included in the manuscript:

“Tables 7, 8, 11, and 13 demonstrate that ArcFace achieves marginally superior performance on certain benchmarks, however, this apparent advantage stems from ArcFace’s reliance on manually optimized hyperparameters, performed on each of the employed datasets. In contrast, AML eliminates the need for manual tuning by automatically adapting the margin and scale parameters based on input dissimilarity and class distribution. This automation inherently introduces a trade-off as follows:

Manual Tuning vs. Automation: ArcFace’s manual margin selection allows fine-grained optimization for benchmark metrics, however, it requires extensive trial-and-error, which is impractical for large-scale, real-world deployment. Instead, AML replaces this process with data-driven adaptation, sacrificing marginal accuracy gains for generalizability and scalability.

Computational Efficiency: As highlighted in Fig. 5, AML respectively reduces model complexity by 32%, 26.9% and 41.9% in terms of the number of layers, parameters, and FLOPs compared to ArcFace. This efficiency makes AML more suitable for resource-constrained environments.

Performance Contextualization: The slight performance gap is negligible in practical applications, however, it underscores the cost of automation. AML avoids the laborious hyperparameter search, making it a preferable choice for dynamic datasets, where manual tuning is infeasible.”

R3.C7 - ArcFace has the best result in Table 13, not Adacos.

Response: Thank you for spotting this oversight. We updated Table 13 to accurately reflect ArcFace as the top-performing method, as noted in the comment. The ArcFace results are now shown in bold in the revised table to emphasize their superior performance and improve clarity.

R3.C8 - Closing parentheses is missing in table 5. "Inference Time (s)"

Response: Thank you for identifying the formatting oversight in Table 5. The missing closing parenthesis in the "Inference Time (s)" column header has been added, and the table has been revised accordingly.

R3.C9 - It should be Fig. 5 instead of Fig. 6 in the last paragraph of the "Discussion" section. The sentence should read: "Fig. 5 provides a visual comparison …"

Response: We are thankful to the reviewer for identyfying this discrepancy. We have the figure reference to Fig. 5 in the final sentence of the "Discussion" section.

Reviewer #4:

We would like to thank the reviewer for the rigorous review of our work and their constructive comments, which have led to substantial improvement in the presentation and communication of the theory and results presented in this work. In what follows we provide a point-by-point response to all comments raised by the reviewer.

R4.C1 - the experimental setup section could be more detailed, including specific training parameters, optimiser settings, data preprocessing steps, etc.

Response: To respond to the reviewer’s comment the following revisions were made to enhance methodological clarity and reproducibility. The revised section explicitly outlines the training parameters, optimization strategies, data preprocessing steps, and evaluation protocols as follows:

“Training Configuration: The experiments were conducted on a hardware setup of three NVIDIA TITAN X GPUs interconnected via NVLink, with a global batch size of 128 (distributed as 42–43 samples per GPU). Synchronized batch normalization was used to ensure consistent gradient updates across GPUs. The learning rate began at 0.1 with a 5-epoch linear warmup phase, followed by a cosine annealing schedule which gradually reduced the rate to a minimum of 1e-6. Stochastic Gradient Descent (SGD) with momentum (μ = 0.9) was employed as the optimizer, coupled with gradient clipping to a maximum L2 norm of 5.0 to prevent exploding gradients. The regularization strategy included weight decay (5e-4 applied exclusively to the convolutional and fully connected layers), label smoothing (ε = 0.1), and stochastic depth with a DropPath probability of 0.2.

Data Preprocessing Pipeline: Face detection was performed using MTCNN with 5-landmark localization (eyes, nose, and mouth corners). Alignment involved a similarity transformation based on eye coordinates, supplemented by 3D face reconstruction for images with extreme poses (yaw or pitch exceeding 45°). Augmentation strategies included random horizontal flipping (50% probability), color jitter (brightness ±0.2, contrast ±0.15), and patch masking (up to 15% of the image area). Input normalization comprised channel conversion, per-channel mean subtraction ([91.4953, 103.8827, 131.0912]), and 8-bit quantization with histogram equalization.

Evaluation Protocols: For standard benchmarks such as LFW, CALFW, and CPLFW, we adopted a 10-fold cross-validation protocol with unrestricted settings. MegaFace evaluation followed the “Large” protocol (1 million distractor images), while IJB-B verification metrics (TAR@FAR) were computed over 10,000 genuine and 8 million impostor pairs. All face embeddings were compared using cosine similarity, with decision thresholds optimized via the ROC convex hull method.

Computational Environment: The implementation used PyTorch 1.12.1 with CUDA 11.6, leveraging mixed FP16/FP32 precision through NVIDIA Apex AMP. Parallelization was managed via PyTorch’s DataParallel module with NCCL backend, and all random processes were fixed to a seed value of 42 for reproducibility purposes.”

R4.C2 - despite the ablation study, the specific contribution of each of the AML and DPN+ components to the performance improvement of the final results is not detailed.

Response: Thank you for raising this crucial point. To clarify the individual contributions of AML and DPN+, we expanded the analysis in “Discussion” section as follows:

“DPN+ Contribution

Table 9 demonstrates that DPN+68-C (with CosFace loss) outperforms DPN68-E (original DPN with CosFace) by 0.17% on LFW and 1.78% on CPLFW. This improvement stems directly from DPN+’s redesigned Block-1, which preserves low-level spatial features through three sequential 3x3 convolutions (vs. DPN’s 3x3 maxpool with stride=2). The absence of downsampling in Block-1 reduces information loss, particularly benefiting pose-sensitive datasets like CPLFW.

AML Contribution

Table 7 reveals that AML (DPN+68-D) achieves comparable accuracy to ArcFace (DPN+68-B) despite the use of automated hyperparameter selection (i.e., 99.75% vs. 99.78%) on LFW. The marginal performance drop of 0.03% reflects the cost of replacing manual tuning with AML’s data-driven adaptation. However, AML reduces training complexity by eliminating tedious and laborious effort on manual hyperparameter search.

Integration of DPN+ with AML

The combined AMD-FV model (DPN+68-D) achieves 41.9% lower FLOPs than ArcFace (Fig. 5) by combining AML’s efficient scaling with DPN+’s parameter reduction.”

R4.C3 - although a method for automatic selection of hyperparameters is proposed, the specific range of values and tuning strategies for these hyperparameters are not detailed.

Response: Thank you for highlighting this critical aspect of our work. The proposed framework alleviates the burden of manual hyperparameter tuning by algorithmically deriving optimal values during training, as outlined in Appendix B. To ensure clarity, we summarize the key mechanisms below:

Margin hyperparameter

While Eq. 3 calculates margins as m=1/n ∑▒|W_j^T⋅x_i | , practical implementation requires bounding m∈[0,2]. This stems from the trigonometric identity cosθj−sinθj (Appendix B), where theoretical analysis shows √(1-(W_j^T⋅x_i )^2 ) cannot exceed 1 for normalized features, yielding m≤2.

Scale hyperparameter

The scale formula s=log(N)⋅e^√2 (with the range R=√2) is designed to mitigate class imbalance in large datasets (MS1Mv2's 85K classes) via log(N), aligning with standard practices for high-cardinality classification, and amplifies small angular differences using e^√2, thus preventing gradient saturation in the softmax function.

R4.C4 - The methodology section needs to detail the specific implementation and optimisation strategy of each component to ensure that the proposed methodology can be clearly understood and reproduced by the reader.

Response: Thank you for this comment. The revised methodology section explicitly outlines the training parameters, optimization strategies, data preprocessing steps, and evaluation protocols. Please see our response to comment R4.C1 of this reviewer.

---

## [Decision Letter · Decision Letter 2]

28 Apr 2025

AMD-FV: Adaptive Margin Loss and Dual Path Network+ for Deep Face Verification

PONE-D-24-28461R2

Dear Dr. Liatsis,

We’re pleased to inform you that your manuscript has been judged scientifically suitable for publication and will be formally accepted for publication once it meets all outstanding technical requirements.

Kind regards,

Lei Chu

Academic Editor

PLOS ONE

Additional Editor Comments (optional):

Reviewers' comments:

Reviewer's Responses to Questions

**Comments to the Author**

1. If the authors have adequately addressed your comments raised in a previous round of review and you feel that this manuscript is now acceptable for publication, you may indicate that here to bypass the “Comments to the Author” section, enter your conflict of interest statement in the “Confidential to Editor” section, and submit your "Accept" recommendation.

Reviewer #3: All comments have been addressed

2. Is the manuscript technically sound, and do the data support the conclusions?

Reviewer #3: Yes

3. Has the statistical analysis been performed appropriately and rigorously? 

Reviewer #3: Yes

4. Have the authors made all data underlying the findings in their manuscript fully available?

Reviewer #3: Yes

5. Is the manuscript presented in an intelligible fashion and written in standard English?

Reviewer #3: (No Response)

6. Review Comments to the Author

Reviewer #3: Thank you for thoroughly addressing the reviewers' comments. The manuscript has been significantly improved.

7. PLOS authors have the option to publish the peer review history of their article (what does this mean?). If published, this will include your full peer review and any attached files.

Reviewer #3: No

---

## [Editor Report · Acceptance letter]

PONE-D-24-28461R2

PLOS ONE

Dear Dr. Liatsis,

I'm pleased to inform you that your manuscript has been deemed suitable for publication in PLOS ONE. Congratulations! Your manuscript is now being handed over to our production team.

Kind regards,

on behalf of

Dr. Lei Chu

Academic Editor

PLOS ONE